# Long-Term Performance of the Magmaris Drug-Eluting Bioresorbable Metallic Scaffold in All-Comers Patients’ Population

**DOI:** 10.3390/jcm11133726

**Published:** 2022-06-28

**Authors:** Arif Al Nooryani, Wael Aboushokka, Bassam AlBaba, Jalal Kerfes, Loai Abudaqa, Amit Bhatia, Anoop Mansoor, Ruwaide Nageeb, Srdjan Aleksandric, Branko Beleslin

**Affiliations:** 1Al Qassimi Hospital, Sharjah 3500, United Arab Emirates; arif345@yahoo.com (A.A.N.); wael.aboushokka@gmail.com (W.A.); albaba@gmail.com (B.A.); jalalkefres@gmail.com (J.K.); loaiabudaqa@gmail.com (L.A.); amitbhatia@doctor.com (A.B.); anoopmansur@gmail.com (A.M.); ruwaida.najeeb@moh.gov.ae (R.N.); 2Faculty of Medicine, University of Belgrade, 11000 Belgrade, Serbia; srdjanaleksandric@gmail.com

**Keywords:** Magmaris, all-comers population, complex patients, long-term outcome

## Abstract

**Background**: The long-term efficacy and safety of bioresorbable vascular scaffolds (BVS) in real world clinical practice including Magmaris need to be elucidated to better understand performance of this new and evolutive technology. The aim of this study was to evaluate long-term performance of Magmaris, drug-eluting bioresorbable metallic scaffold, in all-comers patients’ population. **Methods:** We included in this prospective registry first 54 patients (54 ± 11 years; male 46) treated with Magmaris, with at least 30 months of follow-up. Diabetes mellitus and acute coronary syndrome were present in 33 (61%) and 30 (56%) of the patients, respectively. Patients were followed for device- and patient-oriented cardiac events during a median follow-up of 47 months (DOCE–cardiac death, target vessel myocardial infarction, and target lesion revascularization; POCE–all cause death, any myocardial infarction, any revascularization). **Results:** Event-free survivals for DOCE and POCE were 86.8% and 79.2%, respectively. The rate of DOCE was 7/54 (13%), including in total target vessel myocardial infarction in two patients (4%), target lesion revascularization in six patients (11%), and no cardiac deaths. The rate of POCE was 11/54 (21%), including in total any myocardial infarctions in 3 patients (6%), any revascularization in 11 patients (20%), and no deaths. Definite Magmaris thrombosis occurred in two patients (3.7%), and in-scaffold restenosis developed in five patients (9.3%). Variables associated with DOCE were implantation of ≥2 Magmaris BVS (HR: 5.4; 95%CI: 1.21–24.456; *p* = 0.027) and total length of Magmaris BVS ≥ 40 mm (HR: 6.4; 95%CI: 1.419–28.855; *p* = 0.016), whereas previous PCI was the only independent predictor of POCE (HR: 7.4; 95%CI: 2.216–24.613; *p* = 0.001). **Conclusions:** The results of the long-term clinical outcome following Magmaris implantation in patients with complex clinical and angiographic features were acceptable and promising. Patients with multi-BVS and longer multi-BVS in lesion implantation were associated with worse clinical outcome.

## 1. Introduction

Coronary stents have undergone an impressive evolution over last 30 years, moving from bare metal stents, to overcome vessel recoil, to drug-eluting stents (DES), to further reduce restenosis rate, up to bioresorbable vascular scaffolds (BVS), to eliminate permanent metallic caging and jailing of side branches and provide natural vasomotor activity and endothelial function [1,2]. The majority of the data and clinical trials on BVS have been based on lactate-based polymer systems, including Absorb as the first and the only BVS approved by the FDA [3,4]. In fact, a series of Absorb BVS studies [5,6,7,8] demonstrated promising initial performance and safety, but later analysis of randomized trials [9,10] comparing Absorb BVS with DES demonstrated lower efficacy due to the higher rates of target vessel myocardial infarction and target lesion revascularization. Absorb BVS also had a higher rate of scaffold thrombosis than DES. On the other hand, other BVS technologies based on magnesium alloys, iron, and tyrosine have also been under clinical investigations [11].

Magmaris, a second-generation drug-eluting bioresorbable metallic scaffold that contains magnesium alloy as a backbone, received the EU CE mark in June 2016 and have also shown promising initial clinical results [12,13,14]. The DREAMS 2G scaffold, marketed as Magmaris, was tested in the BIOSOLVE-II and BIOSOLVE-III trials [15,16], including in total 184 stable patients with simple de novo lesions. A pooled analysis of BIOSOLVE-II and BIOSOLVE-III demonstrated a target lesion failure of 5.9% after 24 months and no proven scaffold thrombosis [16]. The BIOSOLVE-IV all-comers registry [17], with more than 1000 patients in Cohort A, showed a similar target lesion failure of 4.3% after 12 months, with a rate of definite or probable scaffold thrombosis of 0.5%.

Following data on the inferior performance of Absorb BVS in comparison with DES, the regulatory medical bodies [18] limited wide application of BVS because of safety issues, whereas a consensus paper recommended restricting the use of Magmaris [19], so the wide application of BVS was interrupted and limited to ongoing clinical studies and registries. Nevertheless, it seems justified for the future evolution of technology to evaluate and analyze the long-term clinical efficacy and safety of BVS in real-world clinical practice, and specifically those of Magmaris because of differences in scaffold technology, including superior mechanical properties [4] and a less thrombogenic profile, to that of the Absorb lactate-based polymer scaffold [20,21]. Thus, the aim of our study was to analyze the efficacy, safety, and long-term clinical outcome of the patients treated in our center with Magmaris, including the all-comers patient population.

## 2. Materials and Methods

### 2.1. Study Population

This was an investigator-initiated prospective registry evaluating the performance, safety, and clinical outcome of Magmaris in the initial 54 patients treated with Magmaris. Patients were enrolled between 2 July 2016 and 17 March 2019, with the vast majority, 43/54 (80%), in 2016 and 2017; 10 in 2018; and only 1 in 2019. The patients included in the study derived from the all-comers patient population referred to PCI who satisfied the eligibility criteria for implantation of Magmaris. Patients were eligible for Magmaris implantation if they were ≥18 years old, including patients with stable angina and evidence of myocardial ischemia on noninvasive imaging as well as patients with acute coronary syndrome. Patients with severe hemodynamic compromise, including cardiogenic shock and/or severe congestive heart failure; patients with limited life expectancy; and patients who could not adhere to prolonged DAPT were not included.

The decision to implant Magmaris was left to the discretion of an operator well experienced in performing standard PCI and trained for Magmaris implantation. The Magmaris BVS is made from a magnesium-alloy scaffold with a strut thickness and width of 120–150 mm. The scaffold is covered by sirolimus in combination with a bioresorbable PLLA polymer (Biotronik AG, Buelach, Switzerland) [13,14,15]. The available Magmaris BVS lengths were 15, 20, and 25 mm, with diameters of 3.0 and 3.5 mm. There were no prespecified limits regarding Magmaris implantation for lesion length, number of target lesions, or number of vessels. Furthermore, in patients with multivessel disease and lesions, there were no limitations in concomitant implantation of DES if the operator considered the lesion not suitable for additional Magmaris implantation. Lesion evaluation was based on a physician’s visual assessment (angio-guided) and scaffold size according to commercially available devices concordant to a reference diameter. Predilatation with a balloon either 0.5 mm smaller than or of equal size to the scaffold device’s diameter was a general strategy. Magmaris was deployed with a slightly lower increase until completely expanded, with optimization including postdilatation with noncompliant balloons. Patients received dual antiplatelet therapy for at least 12 months. Additional OCT or IVUS imaging was recommended and performed during the procedure at the discretion of the operating physician.

All patients who underwent the PCI procedure signed informed consent prior to the procedure according to the hospital policy. The registry was reviewed and approved by the hospital Ethics Committees and a competent authority (MOHAP Research Ethical Committee, approval reference number MOHAP/DXB-REC/AAA/110/2020).

### 2.2. Data Collection, Clinical Follow-Up, and Adverse Events

Patients’ clinical data were prospectively collected and entered into a dedicated database. Procedural data were obtained from catheterization laboratory records and included all relevant information during the PCI. Pre- and postprocedural angiographic characteristics were analyzed by quantitative coronary angiography (QCA) by an independent interventional cardiologist (SA) unaware of the patients clinical or other procedural characteristics. Quantitative QCA data included reference diameter (RD), minimal luminal diameter (MLD), percent diameter stenosis (DS), length of the lesion before and after BVS implantation, and acute gain, defined as the difference between pre- and postprocedural MLD within the BVS.

Clinical follow-up was obtained by telephone contact and/or from national healthcare system medical records in cases where telephone contact was not available. Reported clinical events were checked by medical records and finally verified by an interventional cardiologist (BB) and study nurse (RN) according to criteria for adverse clinical events. There was no systematic or planned repeated angiography following Magmaris implantation—all repeated angiograms were clinically driven and performed only in case of symptoms and signs of myocardial ischemia.

Angiographic success was defined as <30% residual DS by QCA with TIMI 3 flow in the treated target vessel. Procedural success was defined as angiographic success in the absence of in-hospital death, myocardial infarction, or revascularization. Adverse events were classified as device-oriented cardiac events (DOCE), including cardiac death, target vessel myocardial infarction, and target lesion revascularization, and patient-oriented cardiac events (POCE), including all-cause death, any myocardial infarction, and any revascularization. Revascularization was defined as angina- and ischemia-driven consistent with positive functional testing. Myocardial infarction (MI) was defined in accordance with the latest guidelines for MI [22], and stent/scaffold thrombosis (ST) was defined according to ARC criteria [23].

### 2.3. Statistical Analysis

The results are expressed as mean value ± SD or median (interquartile range) depending on the distribution of the data. Dichotomous variables are expressed as percentages. Univariate analysis was used to evaluate the relations between various clinical, procedural, and angiographic variables and the clinical outcome in the follow-up period. To select covariates independently associated with the outcome (DOCE, POCE, or scaffold thrombosis), significant univariable predictors were reassessed by multivariable logistic regression analysis, with values for inclusion and elimination set at *p* < 0.05. Variables entered into the model included all clinical, procedural, and quantitative, semiquantitative, and qualitative angiographic data. Cumulative event rates were based on Kaplan–Meier estimates in time-to-event analysis. Follow-ups for patients were censored on the first event, or in the case of no event, on the last day of the medical contact or available medical records. A *p* value < 0.05 was considered statistically significant. The statistical software package SPSS 26.0 (IBM Corporation, Armonk, NY, USA) was used for statistical analysis. Based on literature data [16], in which the frequency of MACE was 5.9% during 24 months of follow up, the estimated sample size required to estimate the true proportion mean in our study with the required margin of error (5%) and confidence level (95%) was 51.

## 3. Results

### 3.1. Clinical and Angiographic Characteristics

Baseline demographic, clinical, and angiographic characteristics are presented in Table 1. The study included 54 patients (46 males (85%), mean age 54 ± 11 years, range: 27–80 years) with 85 lesions treated with the implantation of 100 stents overall (64 Magmaris BVS and 36 DES). Diabetes mellitus was present in 61% of patients, whereas hypertension, hyperlipidemia, and current smoking history were present in 52%, 52%, and 54% of patients, respectively. Acute coronary syndrome was present in 30 (56%) of patients, including 7 (13%) with STEMI. Previous PCI was performed in 8 (15%), and previous MI was present in 12 (22%), patients. Heart failure with reduced ejection fraction was present in 2 patients, and 1 patient had chronic renal failure. Angiographic and lesion characteristics are presented in Table 2. The one-vessel disease group had 26 (48%) patients, whereas the two- and three-vessel disease groups had 14 patients each. Chronic total occlusion was treated in 2 (4%) patients, bifurcation lesions were treated in 10 (19%) patients, and none of the patients had left main coronary artery stenosis. OCT/IVUS was performed in 43 patients (80%).

### 3.2. Procedural Data and QCA

Procedural characteristics before and after procedure are presented in Table 3. Predilatation of the lesions treated with Magmaris was performed in all patients (100%). The total number of Magmaris BVS implanted was 64 in all patients, with multivessel implantation in 43% of them, 1.3 implanted BVS per lesion, and 1.2 implanted BVS per patient. Additionally, 36 DES were implanted in 46% of patients. The maximum number of implanted Magmaris was four in one patient. Postdilatation of the lesions treated with Magmaris was performed in 53/54 patients (98%), as in 1 patient, hypotension during the procedure developed. Residual dissection following Magmaris implantation was detected in five (8%) lesions, which was treated with additional stenting in four of them. Finally, angiographic and procedural success were obtained in 53/54 patients (98%).

Both percent DS and MLD significantly improved after PCI compared with values before the intervention (57 ± 13% vs. 11 ± 7, *p* < 0.001; 2.85 ± 0.47 vs. 1.22 ± 0.32 mm, *p* < 0.001, respectively).

### 3.3. Clinical Outcome

Median follow-up was 47 months (interquartile range: 34 to 53 months). Complete follow-up was obtained in 53/54 (98%) of the patients. Clinical outcome data are presented in Table 4. DOCE were present in seven patients (13%), including in total target vessel myocardial infarction in two (4%), target lesion revascularization in six (11%), and no cardiac death. On the other hand, POCE developed in 11 patients (21%) including in total all myocardial infarctions in 3 patients (6%), any revascularization in 11 patients (20%), and no deaths. Out of 11 revascularizations in the follow-up period, 6 were not in the target lesions, and one patient had both target and nontarget lesion revascularization. Magmaris thrombosis occurred in two patients (3.7%)—in one patient within 1 month of the intervention, and in the second patient in the 12th month following the intervention. In-scaffold restenosis developed in five patients (9.3%). Event-free survivals for DOCE and POCE were 86.8% and 79.2%, respectively (*p* = NS) (Figure 1).

Variables significantly associated with DOCE were previous PCI (*p* = 0.044), implantation of ≥2 Magmaris BVS (*p* = 0.027), and total length of Magmaris BVS ≥ 40 mm (*p* = 0.016) (Table 5). Two separate multivariate Cox regression analyses (forward method) were performed for all significant univariate predictors for DOCE: one with the implantation of two or more Magmaris stents (Magmaris > 2), and the other with the total Magmaris length > 40 mm implantation per lesion (Models 1 and 2, Table 5). In this manner, the implantation > 2 Magmaris stents and total Magmaris length >40 mm implantation per lesion were independently associated with DOCE during the long-term follow-up (HR: 5.4; 95%CI: 1.21–24.456; *p* = 0.027; HR: 6.4; 95%CI: 1.419–28.855; *p* = 0.016, respectively). These two variables were not taken into account together in multivariate analyses because of their high correlation and multicollinearity (r = 0.916, *p* < 0.001). No interactions were observed in respect to the DOCE and cardiovascular risk factors, clinical presentation, or lesion or procedural characteristics. Figure 2 illustrates a patient with a long and diffuse lesion treated with two Magmaris BVS with diffuse in-scaffold restenosis 6 months later successfully treated with two DES.

Multivariate Cox regression analysis (forward method) showed that previous PCI was the only independent predictor of POCE during the long-term follow-up (HR: 7.4; 95%CI: 2.216–24.613; *p* = 0.001) (Table 6). Double antiplatelet therapy (DAPT) was present in 33/53 (62%) patients at the timepoint of the last follow-up or adverse event and was not related to either DOCE or POCE.

## 4. Discussion

This registry presents the performance, efficacy, safety, and long-term outcomes of the Magmaris BVS in the real-world, all-comers patient population, with significant prevalence of high-risk and complex patients. Having in mind a significant proportion of patients with acute coronary syndrome, diabetes mellitus, and extensive atherosclerotic burden, the Magmaris BVS demonstrated respectable efficacy and safety, with a rate of adverse effects comparable to that in previous studies [15,16,17]. In fact, if the rate of target vessel/lesion failure in the pooled analysis of BIOSOLVE II and III [16] of 5.9% after 24 months (including two cardiac deaths, one target vessel myocardial infarction, four patients with clinically driven target lesion revascularization, and no proven scaffold thrombosis) and that in the large BIOSOLVE IV trial of 4.3% after 12 months (1.1% target-vessel myocardial infarction, 0.2% cardiac death, and 3.9% clinically driven target-lesion revascularization) was interpolated to our data (after median follow-up of almost 4 years, the rate of DOCE was 13%, and that of POCE was 21%), then the long-term outcome seemed not to be affected by the complexity of the patients. On contrary, predictors of DOCE including the number and the length of the Magmaris BVS may imply that future technological innovations of the Magmaris platform may lead to results comparable to those of current DES technology.

Initial experience [5,6,7,8,24,25] with the Absorb BVS demonstrated good procedural safety and angiographic success, as well as short- to medium-term clinical outcome and safety. However, following the AIDA trial [26] showing a few times higher rates of definite and probable scaffold thrombosis in comparison with DES and association with more myocardial infarctions, as well as ABSORB II after 3-year follow-up [27] showing significantly worse outcomes in DOCE for the Absorb BVS in comparison with Xience DES (10% vs. 5%), an FDA safety alert [18] was published, which interrupted commercial application of Absorb and raised the issue of the safety of other BVS.

A recent meta-analysis of randomized trials [10] comparing Absorb BVS with DES demonstrated a higher rate of target lesion failure driven by the higher rates of target vessel myocardial infarction (RR, 1.65; 95% CI, 1.26–2.17) and target lesion revascularization (RR, 1.39; 95% CI, 1.08–1.78). However, the major concern of the Absorb BVS remains the rate of device thrombosis [28]. A meta-analysis of seven randomized trials [29] showed a higher rate of device-oriented adverse events, including significantly higher rates of definite and probable device thrombosis, for the Absorb BVS than for Xience.

However, the Magmaris and Absorb BVS were recently compared in an experimental porcine model [20]. Significantly less scaffold platelet coverage was demonstrated by both Magmaris (3.0%) and Orsiro (4.6%) compared with Absorb (21.8%). Scanning electron microscopy demonstrated significantly less thrombus deposition to Magmaris as a percentage of the total scaffold compared with Absorb (5.0% versus 16.1%, *p* = 0.02). In addition, Magmaris showed significantly less inflammation cell deposit than both Orsiro and Absorb. Waksman et al. [20] concluded that despite a similar scaffold strut thickness, the Magmaris magnesium BVS was significantly less thrombogenic compared with the Absorb BVS in an ex vivo model, suggesting the possibility of less thrombogenic potential in clinical settings or even a potential similar to that of DES.

Finally, the latest data from the real-world practice comparing Magmaris BVS with the novolimus-eluting PLLA-based DESolve scaffold [30] by OCT showed superiority of Magmaris in terms of a larger minimal luminal area, less residual stenosis > 20%, no strut fractures, and more complete scaffold apposition. Thus, the acute mechanical performance of Magmaris seems to be superior to that of DESolve.

### Study Limitations

The limitation of this registry was a relatively small number of patients, which was a consequence of restrictions in implantation of BVS following recommendations on the limited use of BVS [18,19]. However, because of the registry-based profile of the study, the implantation of Magmaris in our center never stopped but occurred at much smaller rates and was indicated only in patients with simple coronary lesions. Still, at this interim phase, data from real-world registries represent an important source of information for the further development of this evolving technology. In addition, because of the limited number of patients and the heterogeneous use of two intracoronary imaging modalities (OCT/IVUS), we did not specifically analyze intracoronary imaging data. The other important limitation of this registry was the absence of systematic coronary imaging in the follow-up; the outcome was directed by clinically driven events. In patients with adverse effects, particularly scaffold thrombosis, the usage of DAPT was based on clinical interview or available medical records and could not be proven with certainty.

## 5. Conclusions

The results of initial real-world Magmaris BVS implantation, even in patients with high-risk clinical and lesion features, were acceptable and corresponded to the results of previous studies with short-term follow-ups and less complex lesions. These data further emphasize the potential of this technology based on magnesium metallic scaffolds. Based on our experience, current application of Magmaris BVS should be limited to 1~2 BVS not exceeding 40 mm.

## Figures and Tables

**Figure 1 jcm-11-03726-f001:**
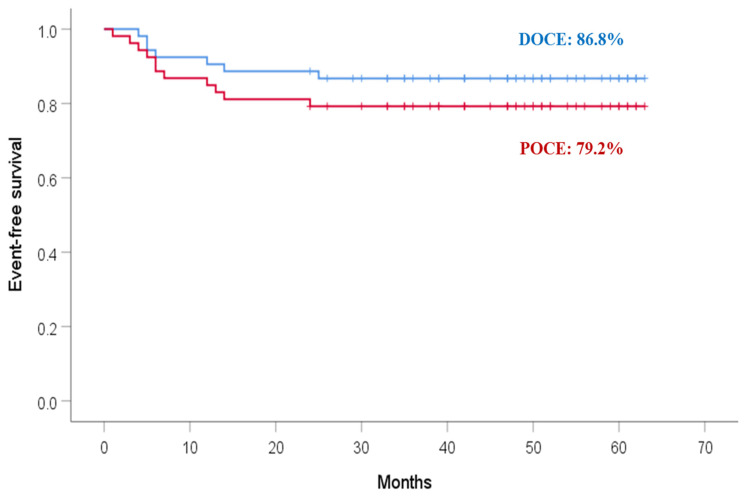
Kaplan–Meier survival analyses during long-term follow-up (median follow-up 47 months) for device-oriented clinical events (DOCE) and patient-oriented clinical events (POCE). DOCE were defined as composite endpoints of cardiac death, target-vessel myocardial infarction, and clinically driven target lesion revascularization. POCE were defined as composite endpoints of all-cause mortality, any myocardial infarction, and any revascularization.

**Figure 2 jcm-11-03726-f002:**
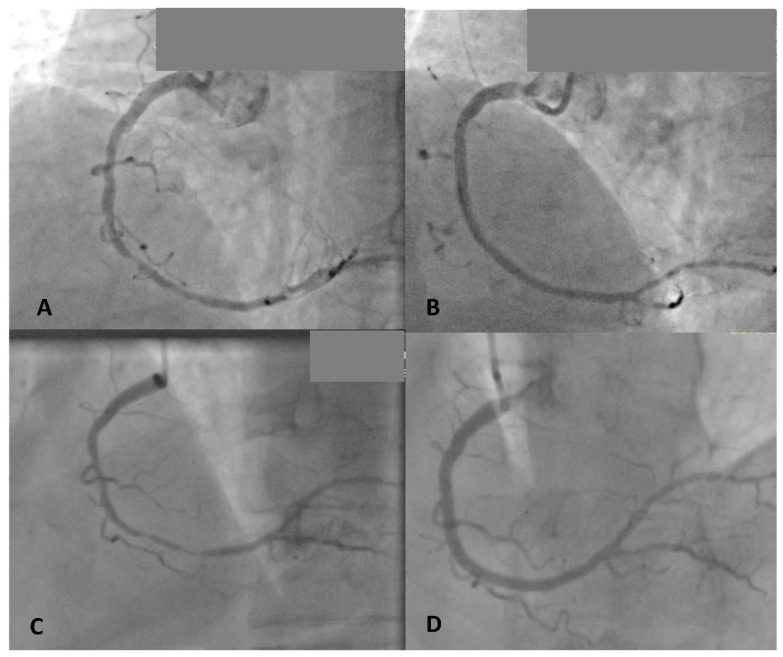
Coronary angiography before and after implantation of 2 Magmaris BVS in the right coronary artery (**A**—diffuse and long stenosis in mid RCA, **B**—after two 3.0 × 25 mm Magmaris implantations) and 6 months later (**C**—diffuse in-scaffold restenosis) with diffuse in-scaffold restenosis treated successfully with 2 DES (**D**).

**Table 1 jcm-11-03726-t001:** Demographic and clinical characteristics of the whole study group.

Variable	All (*n* = 54)
Age ± SD, years	54 ± 11
Gender, males (%)	46 (85)
Previous MI, *n* (%)	12 (22)
Previous PCI, *n* (%)	8 (15)
Previous stroke/TIA, *n* (%)	1 (2)
Hypertension, *n* (%)	28 (52)
Diabetes, *n* (%)	33 (61)
Smoking, *n* (%)	29 (54)
Hyperlipidemia, *n* (%)	28 (52)
Family history, *n* (%)	2 (4)
Heart failure, *n* (%)	2 (4)
Renal failure, *n* (%)	2 (4)
Stable angina, *n* (%)	24 (44)
Unstable angina, *n* (%)	13 (24)
NSTEMI, *n* (%)	10 (19)
STEMI, *n* (%)	7 (13)
Aspirin, *n* (%)	54 (100)
Clopidogrel, *n* (%)	32 (59)
Ticagrelor, *n* (%)	22 (41)

Data are expressed as mean ± SD or as number (%). MI = myocardial infarction; NSTEMI = non-ST-elevation myocardial infarction; PCI = percutaneous coronary intervention; STEMI = ST-elevation myocardial infarction; TIA = transitory ischemic attack.

**Table 2 jcm-11-03726-t002:** Angiographic characteristics.

Number of diseased vessels per patient	
One-vessel disease, *n* (%)	26 (48)
Two-vessel disease, *n* (%)	14 (26)
Three-vessel disease, *n* (%)	14 (26)
Number of treated lesions per patient (*n* = 85)	
One treated lesion, *n* (%)	31 (57)
Two treated lesions, *n* (%)	15 (28)
Three treated lesions, *n* (%)	8 (15)
Number of implanted Magmaris per patient (*n* = 64)	
One stent, *n* (%)	47 (87)
Two stents, *n* (%)	5 (9)
Three stents, *n* (%)	1 (2)
Four stents, *n* (%)	1 (2)
Number of implanted DES per patient (*n* = 36)	
One stent, *n* (%)	17 (31)
Two stents, *n* (%)	5 (9)
Three stents, *n* (%)	3 (6)
Treated vessel	
LAD/DG, *n* (%)	24 (44)
LCx/OM, *n* (%)	9 (17)
RCA, *n* (%)	20 (37)
RIM, *n* (%)	1 (2)
Bifurcation lesion, *n* (%)	10 (19)
Chronic total occlusion, *n* (%)	2 (4)
Access site	
Transradial approach, *n* (%)	52 (96)
Transfemoral approach, *n* (%)	2 (4)

Data are expressed as number (%). DES = drug-eluting stent; DG = diagonal branch; LAD = left anterior descending artery; LCx = left circumflex artery; OM = obtuse (marginal) branch; RIM = ramus intermedius; RCA = right coronary artery.

**Table 3 jcm-11-03726-t003:** Procedural characteristics.

Predilatation, *n* (%)	54 (100)
Predilatation balloon size ± SD, mm	3.12 ± 0.22
Predilatation balloon length ± SD, mm	18.4 ± 3.4
Predilatation balloon pressure ± SD, atm	13.8 ± 2.8
Magmaris diameter ± SD, mm	3.30 ± 0.25
Magmaris length ± SD, mm	21.4 ± 3.7
Magmaris implantation pressure ± SD, atm	13.1 ± 2.3
Postdilatation, *n* (%)	53 (98)
Postdilatation balloon size ± SD, mm	3.54 ± 0.41
Postdilatation balloon length ± SD, mm	12.2 ± 5.2
Postdilatation balloon pressure ± SD, atm	15.6 ± 3.7
Acute gain ± SD, mm	1.23 ± 0.45
Angiographic success (per patient), *n* (%)	53 (98)
Procedural success (per patient), *n* (%)	53 (98)

Data are expressed as mean ± SD or as number (%).

**Table 4 jcm-11-03726-t004:** Clinical outcome data in the whole study group (*n* = 53). One patient was lost in the follow-up.

DOCE, *n* (%)	7 (13)
Cardiac mortality, *n* (%)	0 (0)
Target-vessel myocardial infarction, *n* (%)	2 (4)
Clinically driven target lesion revascularization, *n* (%)	6 (11)
POCE, *n* (%)	11 (21)
All-cause mortality, *n* (%)	0 (0)
All myocardial infarction, *n* (%)	3 (6)
All revacularizations, *n* (%)	11 (20)
CABG, *n* (%)	0 (0)
Nontarget lesion revascularization	6 (11)
In-stent restenosis, *n* (%)	5 (9)
Stent thrombosis, *n* (%)	2 (4)
Acute and early stent thrombosis, *n* (%)	1 (2)
Late stent thrombosis, *n* (%)	1 (2)
Very late stent thrombosis, *n* (%)	0 (0)

Data are expressed as number (%). CABG = coronary artery bypass graft; DOCE = device-oriented clinical events; POCE = patient-oriented clinical events.

**Table 5 jcm-11-03726-t005:** Multivariate Cox regression analyses for all significant univariate variables (*p* ≤ 0.1) predicting device-oriented clinical events (DOCE).

Univariate Analysis	HR (95%CI for OR)	*p*-Value
Age ± SD, years	1.022 (0.954–1.095)	0.536
Gender, males (%)	1.046 (0.126–8.693)	0.967
Previous MI, %	1.398 (0.271–7.213)	0.689
Previous PCI, %	4.657 (1.039–20.880)	0.044
Hypertension, %	2.392 (0.464–12.335)	0.297
Diabetes, %	3.869 (0.466–32.144)	0.210
Smoking, %	2.132 (0.414–10.991)	0.366
Hyperlipidemia, %	5.888 (0.708–48.933)	0.101
Stable angina, %	0.459 (0.127–2.538)	0.459
ACS, %	1.761 (0.394–7.871)	0.459
Bifurcation lesion, %	2.271 (0.508–10.151)	0.283
IVUS and/or OCT, %	0.440 (0.053–3.658)	0.448
LAD/DG, %	0.468 (0.091–2.414)	0.364
LCx/OM, %	2.352 (0.456–12.133)	0.307
RCA, %	1.256 (0.281–5.613)	0.766
Lesion type A, %	0.189 (0.023–1.567)	0.123
Lesion type B1, %	0.956 (0.115–7.940)	0.967
Lesion type B2/C, %	4.037 (0.783–20.811)	0.095
Multivessel PCI, %	0.956 (0.214–4.272)	0.953
Magmaris ≥ 2, %	5.442 (1.211–24.456)	0.027
Magmaris plus DES/BVS, %	1.382 (0.309–6.175)	0.672
Total length of Magmaris ≥ 40 mm, %	6.399 (1.419–28.855)	0.016
**Model 1. Multivariate analysis (forward method)** **with Magmaris ≥ 2**	**HR (95%CI for OR)**	***p*-value**
Magmaris ≥ 2, %	5.442 (1.211–24.456)	0.027
**Model 2. Multivariate analysis (forward method)** **with Total length of Magmaris** **≥ 40 mm**		
Total length of Magmaris ≥ 40 mm, %	6.399 (1.419–28.855)	0.016

Dependent variable: device-oriented clinical events (DOCE). Multivariate Cox regression analyses were adjusted for all variables with *p* ≤ 0.1 in univariate analysis. Variables with frequencies of less than 5 were not placed in the analyses. CI = confidence interval; MI = myocardial infarction; ACS = acute coronary syndrome. IVUS = intravascular ultrasound; OCT = optical coherence tomography; RCA = right coronary artery. Other abbreviations as in Table 1 and Table 2.

**Table 6 jcm-11-03726-t006:** Multivariate Cox regression analyses for all significant univariate variables (*p* ≤ 0.1) predicting patient-oriented clinical events (POCE).

Univariate Analysis	HR (95%CI for OR)	*p*-Value
Age ± SD, years	1.024 (0.970–1.081)	0.388
Gender, males (%)	1.810 (0.232–14.145)	0.552
Previous MI, %	2.235 (0.654–7.637)	0.200
Previous PCI, %	6.168 (1.863–20.421)	0.003
Hypertension, %	1.159 (0.354–3.801)	0.807
Diabetes, %	1.036 (0.303–3.541)	0.955
Smoking, %	1.485 (0.435–5.076)	0.528
Hyperlipidemia, %	2.654 (0.704–10.015)	0.150
Stable angina, %	0.896 (0.273–2.936)	0.856
ACS, %	1.116 (0.341–3.658)	0.856
Bifurcation lesion, %	2.552 (0.778–8.375)	0.122
IVUS and/or OCT, %	0.566 (0.122–2.619)	0.466
LAD/DG, %	1.003 (0.306–3.288)	0.996
LCx/OM, %	1.242 (0.268–5.750)	0.782
RCA, %	0.959 (0.281–3.275)	0.946
Lesion type A, %	0.430 (0.114–1.623)	0.213
Lesion type B1, %	1.810 (0.232–14.145)	0.572
Lesion type B2/C, %	2.843 (0.832–9.717)	0.096
Multivessel PCI, %	1.028 (0.314–3.371)	0.963
Magmaris ≥ 2, %	2.567 (0.679–9.699)	0.165
Magmaris plus DES, %	1.888 (0.552–6.452)	0.311
Total length of Magmaris ≥ 40 mm, %	3.002 (0.793–11.360)	0.105
**Multivariate analysis (forward method)**	**HR (95%CI for OR)**	***p*-value**
Previous PCI, %	7.385 (2.216–24.613)	0.001

Dependent variable: patient-oriented clinical events (POCE). Multivariate Cox regression analysis was adjusted for all variables with *p* ≤ 0.1 in univariate analysis. Variables with frequencies of less than 5 were not placed in the analyses. CI = confidence interval; MI = myocardial infarction; ACS = acute coronary syndrome. Other abbreviations as in Table 1 and Table 2.

## Data Availability

The data presented in this study are available on request from the first author. The data are not publicly available due to privacy issues based on MOHAP administration and ethical standards.

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
