# Peer review of "Long-Term Performance of the Magmaris Drug-Eluting Bioresorbable Metallic Scaffold in All-Comers Patients’ Population"

_jcm, 2022, doi:10.3390/jcm11133726_

Round 1

Reviewer 1 Report

The article «Long-Term Performance of the Magmaris Drug-Eluting Biore-2 sorbable Metallic Scaffold in All-Comers Patients’ Population» is very interesting and the important but provocative conclusions were made. This is a prospective study trying to evaluate the efficacy, safety and long-term clinical outcome of the patients treated in our center with Magmaris, including all-comers patient population.

But I have some comments:

1.     My biggest comment to this article relates to statistics. These observations were made on a relatively small cohort. Please, provide Power calculation

2.     Did you revealed the statistical difference between the rates of DOCE and POCE during follow-up according to Kaplan-Meier survival analyses?

3.     The predictors of DOCE were previous PCI, implantation of ≥ 2 Magmaris BVS, and total length of Magmaris BVS ≥ 40mm, and the authors made the conclusion that Magmaris BVS should be limited to 1-2 Magmaris  BVS not exceeding 40mm. But what was the benefits of Magmaris  BVS implantation?

Author Response

Reviewer 1 comments:

The article «Long-Term Performance of the Magmaris Drug-Eluting Bioresorbable Metallic Scaffold in All-Comers Patients’ Population» is very interesting and the important but provocative conclusions were made. This is a prospective study trying to evaluate the efficacy, safety and long-term clinical outcome of the patients treated in our center with Magmaris, including all-comers patient population.

We would like to thank the Reviewer for important comments that improved the quality of the manuscript and clarified some important aspects of the study.

Comments to the questions:

  1. My biggest comment to this article relates to statistics. These observations were made on a relatively small cohort. Please, provide Power calculation

Thank you for this important comment that might further improve reliability of our study. We acknowledged this issue in the first sentence of the Study Limitations confirming that this small number of patients is the major limitation of the study which appeared to be an unavoidable consequence of the general restriction in implantation of bioresorbable vascular scaffolds following data on Absorb BVS.  However, based on the calculation of the sample size we have added following sentence on the power calculation which is based on previous published data on Magmaris adverse effects over 24 months of 5.9% (end of Statistical analysis, row 141). Based on literature data (16), where the frequency of MACE was 5.9% during 24 months of follow up, the estimated sample size required to estimate the true proportion mean in our study with the required margin of error (5%) and confidence level (95%) was 51.

  1. Did you reveal the statistical difference between the rates of DOCE and POCE during follow-up according to Kaplan-Meier survival analyses?

Thank you for this important point that might further emphasize the complexity of the patients. Although there was more patients’ than device oriented adverse events, this difference did not reach statistical significance. So we have added statistically nonsignificant difference in the Results section – Clinical outcome, row 195. 

  1. The predictors of DOCE were previous PCI, implantation of ≥ 2 Magmaris BVS, and total length of Magmaris BVS ≥ 40mm, and the authors made the conclusion that Magmaris BVS should be limited to 1-2 Magmaris  BVS not exceeding 40mm. But what was the benefits of Magmaris  BVS implantation?

Thank you again for this critical and sensitive issue. At this interregnum phase of BVS technology, it seems prudent to understand limitations of new technology, and acknowledge potential benefits of BVS implantation which is currently challenged by the restricted use of BVS to only on-going registries and clinical trials. So, to our opinion at this timepoint it seems reasonable to be modest in the conclusions, careful in benefits and stick to the evidence and observations demonstrated in our study.         

Reviewer 2 Report

The current paper from Arif Al Nooryani et al  studies the long term performance of Magmaris as a BVS.

The paper is well written and this real-world registry was conducted with a good methodology.

I have the following queries:

  • Patients enrollment was stopped in 2019 with only 54 patients enrolled. Why? Please address this point in the limitations 
  • Are data about therapeutic adherence (DAPT in particular) available?
  • Authors report “The results are expressed as mean value ± SD, or median (interquartile range) depending on the distribution of the data.” Only follow up period was reported as median and interquartile range. Were other variables tested for normal distribution? It is a bit strange that every variable was normally distributed
  • Authors report “OCT/IVUS was performed in 43 patients (80%).” No data about intracoronary imaging was reported, why?
  • Please add statistical software used
  • Please use clopidogrel instead of plavix

Author Response

Reviewer 2 comments:

The current paper from Arif Al Nooryani et al  studies the long term performance of Magmaris as a BVS.

The paper is well written and this real-world registry was conducted with a good methodology.

We would like to thank Reviewer 2 for the kind comments on our manuscript and carefully address all raised issues.

I have the following queries:

Patients enrollment was stopped in 2019 with only 54 patients enrolled. Why? Please address this point in the limitations.

Thank you for this crucial issue that we have specifically and in detail emphasize in Methodology section.  In fact, wide application of BVS was interrupted and limited to ongoing clinical studies and registries following data of the inferior performance of Absorb in comparison to DES. So, in the light of ongoing registry, application of Magmaris never stopped in our center, but the rate of implantation was much smaller. In addition, as the focus of this study was evaluation of  long-term performance, we have included only patients with at least 30 months of follow-up, fulfilling power calculation of the sample size which was also added in Statistical analysis (row 141). According to your suggestions, we have extended the first sentence of Study limitations as following:

The limitation of this registry is a relatively small number of patients, which is the consequence of restrictions in implantation of BVS following recommendation on limited use of BVS (18,19). However, due to the registry-based profile of the study, the implantation of Magmaris in our center never stopped but occurred at much smaller rate and was indicated only in patients with simple coronary lesions.

Are data about therapeutic adherence (DAPT in particular) available?

Thank you for this important note, which was recognized and commented in Study limitations (last sentence in Study limitations). Regarding your specific question, we have added following sentence at the end of Results section (row 232): DAPT was present in 33/53 (62%) patients at the time-point of the last follow-up or adverse event, and was not related to either DOCE or POCE.

Authors report “The results are expressed as mean value ± SD, or median (interquartile range) depending on the distribution of the data.” Only follow up period was reported as median and interquartile range. Were other variables tested for normal distribution? It is a bit strange that every variable was normally distributed

Indeed, for all variables the normality of distribution was tested first and analyzed correspondently.

Authors report “OCT/IVUS was performed in 43 patients (80%).” No data about intracoronary imaging was reported, why?

Thank you for this comment! IVUS/OCT imaging was recommended, but not mandatory, and the number reflects the nature of the registry and true clinical practice. Also, during this initial worldwide phase of Magmaris implantation, the overall benefits of intracoronary imaging were not fully appreciated. In addition, due to the limited number of patients and heterogenic use of 2 intracoronary imaging modalities (OCT/IVUS) we have not specifically analyzed intracoronary imaging data.  However, we consider your observation important and we have added this sentence in Study Limitations (row 299).

Please add statistical software used

We have used SPSS 26.0(IBM Corporation, Armonk, New York), and we have added this information in Statistical analysis (row 139).

Please use clopidogrel instead of Plavix

Thank you for this observation, we have changed Plavix to clopidogrel in Table 1.

Round 2

Reviewer 1 Report

I’d like to thank the authors because they answered all my questions and provided all corrections 

Reviewer 2 Report

Thank you to the authors for better addressing the issues. I think that this paper reached a good level with this revision.